# Unicompartmental Knee Arthroplasty for Osteoarthritis Eliminates Lateral Thrust: Associations between Lateral Thrust Detected by Inertial Measurement Units and Clinical Outcomes

**DOI:** 10.3390/s24072019

**Published:** 2024-03-22

**Authors:** Hikaru Sato, Hiroaki Kijima, Takehiro Iwami, Hiroaki Tsukamoto, Hidetomo Saito, Daisuke Kudo, Ryota Kimura, Yuji Kasukawa, Naohisa Miyakoshi

**Affiliations:** 1Department of Orthopedic Surgery, Akita University Graduate School of Medicine, 1-1-1 Hondo, Akita 010-8543, Japan; 2Department of System Design Engineering, Faculty of Engineering Science, Akita University Graduate School of Engineering Science, 1-1 Tegatagakuenmachi, Akita 010-8502, Japan; 3Noshiro Kousei Medical Center, Ochiaiazakamimaedachinai, Noshiro 016-0014, Japan; 4Division of Rehabilitation Medicine, Akita University Hospital, 44-2, Hiroomote Hasunuma, Akita 010-8543, Japan

**Keywords:** knee osteoarthritis, inertial measurement sensor units, unicompartmental knee arthroplasty, lateral thrust

## Abstract

The purpose of this study was to investigate the relationship between clinical outcomes and lateral thrust before and after unicompartmental knee arthroplasty (UKA) using inertial measurement sensor units. Eleven knees were evaluated with gait analysis. The varus angular velocity was used to evaluate lateral thrust. The femorotibial angle (FTA) and hip–knee–ankle angle (HKA) were used to evaluate lower-limb alignment, and the Oxford Knee Score (OKS) and Japanese Orthopaedic Association Score (JOA) were used to evaluate clinical outcomes. The mean pre-UKA peak varus velocity was 37.1 ± 9.8°/s, and that for post-UKA was 28.8 ± 9.1°/s (*p* = 0.00003), such that instabilities clearly improved. Assuming the definition of lateral thrust is when the varus angular velocity is more than 28.1°/s, 81.8% of patients had lateral thrust preoperatively, but this decreased to 55.6% postoperatively, such that the symptoms and objective findings improved. Both OKS and JOA improved after surgery. In addition, HKA was −7.9° preoperatively and −5.8° postoperatively (*p* = 0.024), and FTA was 181.4° preoperatively and 178.4° postoperatively (*p* = 0.012). There was a positive correlation between postoperative JOA and FTA, indicating that changes in postoperative alignment affected clinical outcomes. This study quantitatively evaluated the disappearance of lateral thrust by UKA, and it found that the stability can be achieved by UKA for unstable knees with lateral thrust.

## 1. Introduction

Osteoarthritis of the knee (knee OA) increases with age and has become a social problem, although only 1/2 to 1/3 of radiographic osteoarthritis cases are reported to be symptomatic [1]. It is estimated that there are about 25 million people with radiographic knee OA and about 8 million symptomatic patients in Japan [2].

The Research on Osteoarthritis/Osteoporosis Against Disability (ROAD) study reported that the prevalence of radiographic knee OA increases rapidly from the age of 50 years, with more than 40% of men and 60% of women in their 70s having Kellgren–Lawrence grade 2 or higher knee OA [1]. This trend is similar worldwide, with 35–42% of women in a North American cohort study of residents aged 60 years and older having radiographic knee OA [3]. Reduced mobility due to knee pain caused by this disease can affect healthy life expectancy and prognosis, making more effective treatment of knee OA pain a global challenge.

When conservative treatments fail to improve knee OA, surgical intervention becomes the chosen treatment. This can take the form of total knee arthroplasty (TKA), unicompartmental knee arthroplasty (UKA), or around-knee osteotomy (AKO), a procedure that preserves the joint. TKA involves replacing the entire femorotibial joint with a prosthesis, addressing not only issues related to cartilage wear and meniscal degeneration but also problems with alignment and stability of the entire lower limb caused by bone damage and deformation. Despite the effectiveness of TKA, some patients express dissatisfaction due to discomfort and a restricted range of motion resulting from the placement of a large metal implant in a relatively shallow location.

In contrast, AKO preserves not only the patient’s own bone and cartilage but also joint proprioception. This is achieved by correcting the alignment to allow the remaining cartilage to bear weight and by repositioning the bone to reduce instability. The range of motion is almost unrestricted. Moreover, because AKO does not carry the risk of implant breakage or peri-prosthetic infection, it has a high satisfaction rate among active patients who wish to continue manual labor or outdoor sports activities postoperatively. However, one drawback of AKO is that it does not alleviate the pain associated with cartilage wear and meniscal degeneration.

The UKA procedure is a surgical procedure that replaces only worn cartilage and deformed bone due to cartilage wear and meniscal degeneration, does not require the large metal implant of TKA, and preserves soft tissues such as ligaments so that joint-specific sensation is maintained, and there is little postoperative discomfort or limited range of motion. However, because it does not aim to correct alignment of the entire lower extremity or restore stability, its long-term results are inferior to those of TKA [4], and a high revision rate has been reported, especially in younger patients [5].

Recently, however, it has been suggested that, with improved implants and precise surgical techniques, UKA can be performed for relatively severe unicompartmental deformities, and even if only worn cartilage and bone deformed by cartilage wear are replaced, alignment can be improved and stability can be restored. In fact, research to date has reported that UKA is associated with less postoperative pain than TKA [6], a higher rate of improvement in the Oxford Knee Score (OKS), and a higher rate of return to sports, with functional assessments being comparable or better [7,8].

In particular, there have been reports of improvement in lower-extremity alignment, even in UKA, and an association between improved alignment and clinical outcomes [9]. However, there are still no quantitative reports on whether stability is regained as a result of alignment changes. In particular, there are no reports of how many patients with severe instability, such as lateral thrust, are candidates for UKA surgery, whether lateral thrust resolves after UKA surgery, or whether the course of lateral thrust from before to after UKA is associated with clinical outcomes.

Gait analysis is necessary to examine such issues. In state-of-the-art research, it has become possible to automatically classify gait patterns using the metaheuristic optimization algorithm model [10]. Similarly, deep neural network models and layer-wise relevance propagation technologies have been applied to gait analysis in recent years [11]. In addition to these latest technologies, there have been many useful studies in which gait analysis was conducted using methods such as filming subjects walking over a force platform [12].

Lateral thrust is a phenomenon in which the knee suddenly moves outward in the early stance phase as knee OA progresses, and it is known to increase as knee OA progresses [13]. Lateral thrust has been reported to be strongly associated with pain, and a decrease in lateral thrust is thought to lead to an improvement in pain [14].

Previously, lateral thrust was assessed either visually or through camera images. However, visual inspection only allows for a binary evaluation of lateral thrust: it is either present or absent, and this assessment can vary among different examiners. Moreover, motion capture evaluations are unable to capture intricate movements like lateral thrusts due to the limitations of camera performance. In contrast, inertial sensors can accurately record minute movements, such as lateral thrusts, in the form of acceleration. Tsukamoto et al. were able to quantify lateral thrust in patients with knee OA using inertial measurement sensor units (IMUs) [15]. Using this method, if the change in lateral thrust can be quantitatively evaluated before and after UKA using IMUs, it will be possible to not only clarify that UKA can be performed for patients with instability that causes lateral thrust but also quantitatively show that lateral thrust improves after UKA surgery. If it can be quantitatively confirmed that lateral thrust improves after UKA, the possibility of poor long-term outcomes due to residual instability after UKA can be ruled out, and this could dramatically change the way of thinking about surgical indications for knee OA, which has been a global problem.

Therefore, the purpose of this study was to measure changes in lateral thrust before and after UKA using IMUs and to investigate the relationship with clinical outcomes.

## 2. Materials and Methods

### 2.1. Patients

Of all patients who visited our university hospital and underwent UKA for the diagnosis of knee osteoarthritis, all patients who were able to undergo gait analysis with IMUs between June 2022 and November 2023, both immediately before UKA and at the follow-up visit more than 6 months after UKA, were included in this study. All of these patients were able to walk 10 m without a cane, none had a walking disability due to stroke or neurological disease, and none had psychiatric disorders. Approval for this study was granted by the institutional review board of our university, and the patients provided informed consent to participate in this study.

### 2.2. Gait Analysis

Gait analysis was performed twice: before UKA surgery for knee OA and at least six months after surgery, when walking ability and muscle strength had reached a certain level. For gait analysis, IMUs (IMU-Z2, ZMP Inc., Tokyo, Japan) were attached to the waist, both thighs, and both lower legs with straps (Figure 1). As in previous studies, the sampling rate was 100 Hz [15].

The subject walked 10 m three times while wearing an inertial sensor, and the average data were used. The subject started in an upright position. The start and end of the walk were signaled, during which the subject walked as usual. A video camera was used to record the participants while they walked, and the recording was used to evaluate their walking speed. The data measured by the inertial sensor were synchronized and recorded on a laptop (LAVIE Hybrid ZERO; NEC Corporation, Tokyo, Japan) via Bluetooth. As in the previous study, the internal and external angles of the knee joint were calculated using an extended Kalman filter, and the angular velocity of the knee was calculated by differentiating the calculated angles [16]. The cutoff value for the presence of lateral thrust was the knee internal rotation angular velocity of 28.1°/s obtained in a previous study [15]. The study participants were asked to walk at their own comfortable pace along a 10 m walkway. This gait analysis was conducted not just for research but also for potential clinical applications. Therefore, the gait speed was not standardized, allowing for the replication of each individual’s natural walking rhythm. The subjects performed this task on a wooden floor, wearing their shoes. To ensure stability during the walk, a buffer zone of at least three steps was provided both before and after the 10 m walkway. The participants equipped with IMUs underwent three gait tests. The average value derived from these three tests was used to represent a single data point for each subject, in line with the methodology of previous studies. To familiarize themselves with the motion of walking while wearing the sensors, the subjects were asked to walk several times in a straightforward direction at a speed they found comfortable. Moreover, to mitigate the potential impact of fatigue on the test results, a standing rest period of approximately 10 s was provided between each of the three walking tests. This ensured that each test was conducted under similar conditions, thereby enhancing the reliability of the results.

### 2.3. Radiographic Analysis and Clinical Outcome Assessment

Lower-extremity alignment was assessed on full-length standing frontal radiographs of the lower extremities using the femorotibial angle (FTA), which is the angle between the femoral axis and the tibial axis, and the hip–knee–ankle angle (HKA), which is the angle between the center of the femoral head and the center of the distal femoral intercondylar ridge and the center of the ankle joint. In other words, the FTA is a measure of the angle formed by the femur and tibia at the knee joint. When this angle exceeds 180 degrees, it signifies an increase in varus deformity, which is typically caused by wear on the medial compartment of the knee. Thus, the FTA serves as an anatomical indicator of this condition [17,18]. The HKA is a measure of mechanical alignment, indicating the deviation of the knee joint from the load-bearing axis defined by the hip and ankle. An increase in knee varus during load corresponds to a larger HKA, suggesting greater mechanical stress on the medial compartment of the knee [19,20].

Clinical outcomes were assessed using the OKS patient rating scale and the Japanese Orthopaedic Association Score (JOA score), which assesses pain and ability to walk, pain and ability to climb stairs, range of motion, and swelling.

### 2.4. Statistical Analysis

Statistical analysis was performed using EZR (Ver. 1.61) [21]. Spearman’s rank correlation coefficient was used to describe the relationships between lower-extremity alignment and clinical outcomes and between lateral thrust and clinical outcomes, and the Wilcoxon signed rank sum test was used to compare OKS and JOA scores before and after surgery.

## 3. Results

The study included 11 knees of eight patients. The mean age of the patients was 75 years (ranging from 64 to 82 years). Four patients were male and four were female. The average height of the subjects was 157.4 ± 7.06 cm. The patients’ weight was 62.6 ± 7.94 kg at the time of gait analysis immediately before UKA, and 62.8 ± 9.58 kg at the time of gait analysis more than 6 months after UKA (paired *t*-test, *p* = 0.065).

Postoperative measurements were taken more than 6 months after the patients’ muscle strength had fully recovered, and it was thought that walking was stable. However, due to individual patient circumstances, some cases were measured after a relatively long period of time, resulting in an average follow-up period of 9.1 months.

In a previously reported study using the same IMUs and performing the same gait analysis, the cutoff value for peak knee varus angular velocities with visualized lateral thrust was 28.1°/s on the receiver operating characteristic (ROC) curve. At this cutoff value, the sensitivity was 0.957, and the specificity was 0.579. Therefore, in the present study as well, a peak knee varus angular velocity of 28.1°/s or higher was considered lateral thrust. The pre-UKA peak varus velocity of the knee was higher than the cutoff value for lateral thrust (28.1°/s) in nine of eleven knees (mean 37.1°/s, standard deviation 9.8°/s), such that 81.8% of the pre-UKA knee had lateral thrust. On the other hand, the mean peak varus velocity after UKA was 28.8°/s, with a standard deviation of 9.1°/s, and peak varus velocity was significantly improved by UKA (Figure 2). Only 4 of the 9 knees that had lateral thrust pre-UKA had lateral thrust after UKA. Thus, UKA reduced the lateral thrust cases by 55.6%.

The age, sex, height, weight, and walking speed of the subjects are shown in Table 1. Although UKA reduced pain and increased walking speed in all cases, angular velocity and lateral thrust were significantly reduced. As the walking speed increases, the angular velocity also increases, which may result in excessive lateral thrust being detected, but the angular velocity decreased, confirming quantitatively that UKA improves knee stability.

Regarding clinical outcomes, the OKS was 28.3 preoperatively and 40.1 postoperatively (*p* = 0.003), and the JOA score was 62.5 preoperatively and 89.1 postoperatively (*p* = 0.003) (Figure 3). The patients originally requested surgery due to severe knee pain, but after undergoing UKA, their pain improved dramatically, making it easier to carry out activities of daily living, and their quality of life also improved. The fact that the clinical score improved significantly reflects the above results.

The results for lower-limb alignment were similar to the clinical outcome results. The HKA was −7.9° preoperatively and −5.8° postoperatively (*p* = 0.024), and the FTA was 181.4° preoperatively and 178.4° postoperatively (*p* = 0.012) (Figure 4).

Preoperatively, there were no correlations between the OKS/JOA scores and the HKA/FTA. On the other hand, postoperatively, a positive correlation was observed between the JOA scores and the FTA (0.682, *p* = 0.02) (Figure 5).

## 4. Discussion

In the present study, whether lateral thrust can be eliminated by replacing only the deformed bone with a bone graft was investigated. Lateral thrust was quantified using IMUs, and more than 80% of preoperative UKA patients were found to have lateral thrust. Moreover, to the best of our knowledge, this is the first study to show that UKA decreased the lateral thrust cases by 55.6%.

UKA is a replacement of only the worn part of one compartment, so unlike TKA, which replaces the entire knee, UKA is not a procedure to correct the alignment of the lower limb. In addition, it cannot be combined with osteotomy to improve instability, as is the case with AKO, so it cannot aim to restore stability, and it has been said that long-term results are inferior to those of TKA, and the revision rate is high [4,5]. However, the results of the present study show that, even when performed for unstable knees with lateral thrust, the procedure can restore stability and eliminate lateral thrust due to recent improvements in implants and in surgical technique.

In evaluating lateral thrust, most previous studies have only assessed the presence or absence of lateral thrust visually by the examiner or attempted to capture slight changes using optical motion capture [22,23,24]. However, this study quantitatively evaluated the degree of lateral thrust by using IMUs that can quickly and accurately detect small movements, and thus, it was possible to quantitatively demonstrate that stability was significantly improved by UKA.

Although the phenomenon of lateral thrust caused by knee instability associated with knee OA is a clinically very important finding, it has been only evaluated visually by doctors or by analysis using images from optical cameras. However, the examiner’s visual inspection can only assess the presence or absence of lateral thrust, and even the determination of the presence or absence is ambiguous due to the examiner’s subjectivity. In addition, even with motion capture evaluation, there are limits to camera performance, so it is not possible to capture fast, detailed movements like lateral thrusts. However, by using IMUs, it is possible to accurately capture fast, detailed movements. Moreover, IMUs are small and inexpensive, and do not require large-scale equipment like optical motion capture, making them extremely useful for research like the present study, but they have not yet received much attention worldwide. In addition, the accuracy of IMUs has improved markedly in recent years, and we believe that IMUs will be used in many studies in the future. The accuracy of the IMUs used in the present research was also at the highest level, able to measure angular velocity extremely accurately.

Furthermore, the degree of improvement in lateral thrust correlated with the degree of improvement in clinical outcomes, and improvement in lower-limb alignment was also observed. This suggests that UKA not only relieves pain associated with cartilage wear and meniscal degeneration, but it also improves alignment of the entire lower extremity and achieves overall knee stability, thereby contributing to symptom improvement in patients with knee OA.

TKA is a very effective procedure that can treat overall lower-extremity malalignment and overall knee instability due to deformity, but it is highly invasive and unsatisfactory due to discomfort and limited range of motion caused by loss of joint-specific sensation. The results of the present study confirm that UKA can be performed for unstable knees with unicompartmental arthropathy, such as those with lateral thrust, such that symptoms associated with cartilage wear and meniscal degeneration can be virtually eliminated with UKA, and even stability can be regained with little postoperative discomfort and limited range of motion because soft tissues can be preserved.

However, a limitation of this study is the small number of cases, because the IMU assessment method is not yet a test that can be used clinically in general facilities. In addition, one of the limitations of this study is that the study did not collect the target number of patients according to the calculated required sample size. Thus, the small sample size is one of the biggest concerns about this study. However, it is research that used sensors and methods that have already been established in previous studies [25], and despite the small sample size, there were significant results. The results obtained can be said to be quantitative evidence of clinical relevance.

Another limitation of this study is that the possibility of recurrence of instability and the need for revision surgery after a longer period of time could not be assessed because the evaluation was performed only during the six-month postoperative period.

Furthermore, an additional limitation of this study is that it was not possible to clarify the differences between cases in which lateral thrust improved and cases in which it did not. We believe that, by creating a system that can routinely perform preoperative and postoperative evaluations using IMUs, we will be able to increase the number of cases, elucidate the differences between the above two groups, and report on them to be useful in the treatment of knee OA in the future.

If IMUs could be used very easily to confirm stability in the perioperative period, they could be used not only preoperatively but also frequently, such as 1 month, 3 months, and 6 months postoperatively. Since angular velocity testing can be performed frequently, it will likely be possible to investigate the time it takes to achieve stability. Although this research is limited in that it has not yet been able to clarify this point, its results will lead to future research.

As sensor technology advances and IMUs become more widely used to evaluate lateral thrust, it will be possible to evaluate many more patients worldwide in the same manner as in the present study. When this happens, surgical treatment of knee OA, especially UKA, will expand, and the method will be able to help more patients with knee OA. This study provides important preliminary evidence for such future large-scale clinical trials and future advances in the treatment of knee OA.

## 5. Conclusions

The results of the present quantitative evaluation of lateral thrust using IMUs showed that lateral thrust was observed in 81.8% of patients who underwent UKA, and that UKA decreased lateral thrust cases by 55.6%.

Lower-extremity alignment was also significantly improved after UKA compared to preoperatively.

Improvement in lower-extremity alignment was associated with better postoperative clinical outcomes.

## Figures and Tables

**Figure 1 sensors-24-02019-f001:**
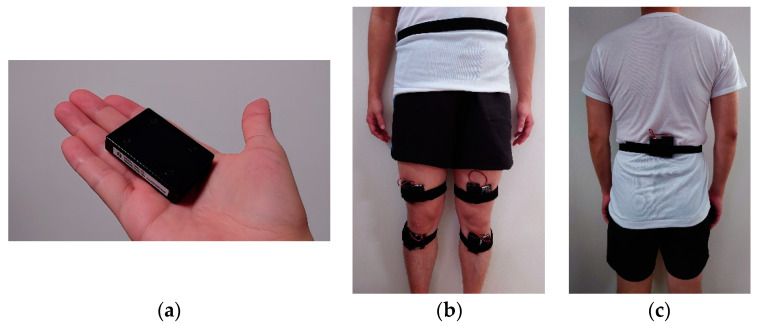
Gait was analyzed using 9-axis inertial measurement sensor units (IMUs) equipped with a 3-axis accelerometer, a gyro sensor, and a magnetometer (**a**). The IMUs are attached to the lower back, both thighs, and both lower legs using a band (**b**,**c**).

**Figure 2 sensors-24-02019-f002:**
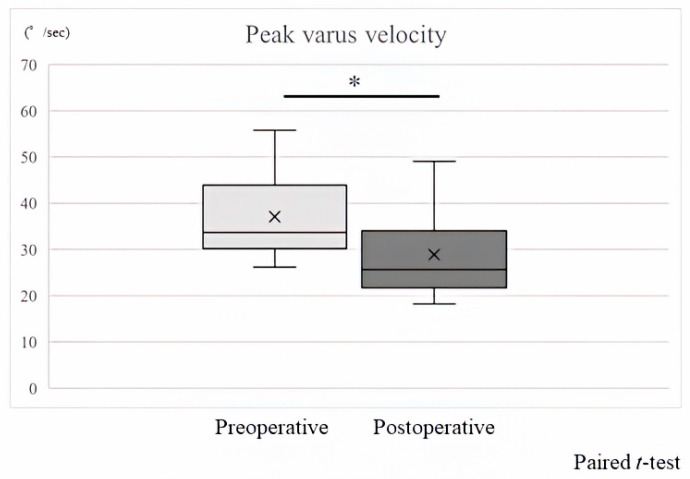
The mean pre-UKA peak varus velocity of the knee is 37.1°/s, with a standard deviation of 9.8°/s, and the mean peak varus velocity after UKA is 28.8°/s, with a standard deviation of 9.1°/s (*p* < 0.05). * Significant difference between Preoperative versus Postoperative (*p* < 0.05).

**Figure 3 sensors-24-02019-f003:**
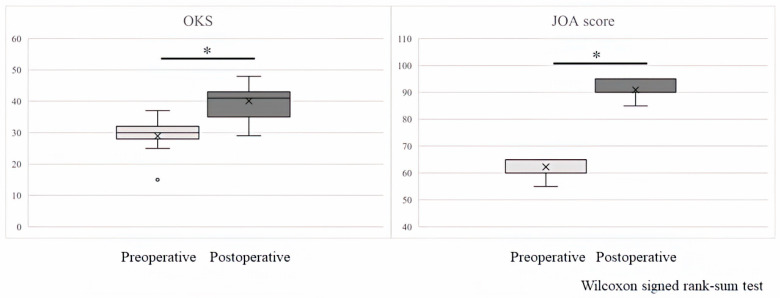
The OKS is 28.3 preoperatively and 40.1 postoperatively (*p* = 0.003), and the JOA score is 62.5 preoperatively and 89.1 postoperatively (*p* = 0.003). * Significant difference between Preoperative versus Postoperative (*p* < 0.05).

**Figure 4 sensors-24-02019-f004:**
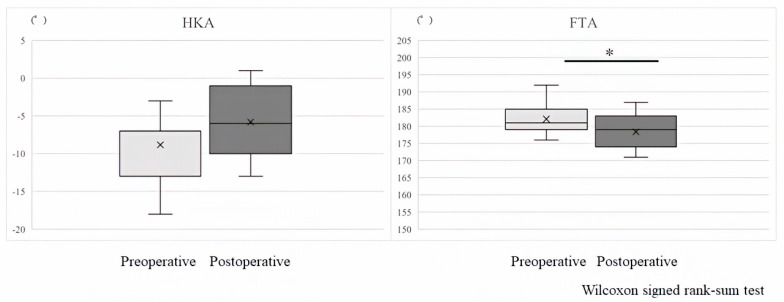
The HKA is −7.9° preoperatively and −5.8° postoperatively (*p* = 0.024), and the FTA is 181.4° preoperatively and 178.4° postoperatively (*p* = 0.012). * Significant difference between Preoperative versus Postoperative (*p* < 0.05).

**Figure 5 sensors-24-02019-f005:**
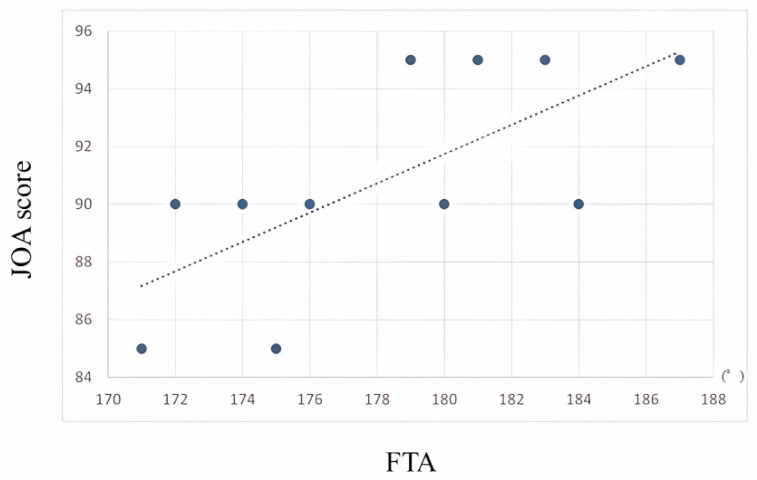
There is a significant positive correlation (0.682, *p* < 0.05) between the JOA score and the FTA.

**Table 1 sensors-24-02019-t001:** Background of the subjects.

Patient No.	Age (Years)	Sex	Gait Speed (m/s) Preoperative/Postoperative	Height (cm)	Body Weight (kg) Preoperative/Postoperative
1	75	Female	0.76/0.77	146.0	65.5/71.0
2	70	Female	1.09/1.14	154.8	56.1/54.0
3	80	Male	0.70/0.73	159.2	58.7/57.4
4	75	Female	0.80/1.23	154.0	50.8/51.0
5	82	Male	0.68/0.86	161.7	61.9/62.5
6	64	Male	0.94/1.11	158.1	79.3/81.1
7	74	Male	0.68/0.80	171.3	63.5/64.0
8	75	Female	0.77/1.03	155.4	65.3/60.0

## Data Availability

Our data are not available due to privacy and ethical restrictions.

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
