# Peer review of "Unicompartmental Knee Arthroplasty for Osteoarthritis Eliminates Lateral Thrust: Associations between Lateral Thrust Detected by Inertial Measurement Units and Clinical Outcomes"

_sensors, 2024, doi:10.3390/s24072019_

Round 1

Reviewer 1 Report (Previous Reviewer 1)

Comments and Suggestions for Authors

The quality of the manuscript has improved somewhat after revisions. However, there are still certain issues that need to be modified before being accepted.

1. The reviewer suggests that consider adding a sentence or two in the abstract about the clinical implications of the findings.

2. From the reviewer's perspective, while the clarity of Figure 1 has improved, the overall composition framework could be made more aesthetically pleasing.

3. The reviewer suggests that the author provide an explanation of how the sample size for this study was calculated and on what basis.

4. Lines 150-161The reviewer suggests that the author include additional references to underscore the rigor and persuasiveness of the study in this section

5. This issue is of great importance: gait analysis is an important part of the current study (as shown by the contents of the manuscript), but nothing in the introduction is related to this. The reviewers were confused by this, which is not a normal state of affairs for a research paper. So, please add a section on gait analysis in the introduction section. The authors may consider citing the following state-of-the-art research to argue the point about gait analysis: (1) A new method proposed for realizing human gait pattern recognition: inspirations for the application of sports and clinical gait analysis; (2) Explaining the differences of gait patterns between high and low-mileage runners with machine learning; (3) Foot Morphology and Running Gait Pattern between the Left and Right Limbs in Recreational Runners.

6. At present, the reviewer expresses a heightened concern regarding the inherent limitations of this study. This apprehension stems from the relatively modest sample size and the somewhat constrained precision associated with the Inertial Measurement Unit (IMU). The accuracy of IMU measurements is not particularly elevated, and the study's reliance on a limited number of samples prompts inquiries into the broader applicability and robustness of the findings.

The reviewer needs to reconsider the acceptability of the manuscript before the authors address these issues well.

Comments on the Quality of English Language

no

Author Response

Reviewer 2 Report (Previous Reviewer 2)

Comments and Suggestions for Authors

The revised paper has been rewritten to better reflect the reviewers' points and meets the standards for publication in the journal, Sensors.

Author Response

Reviewer 3 Report (New Reviewer)

Comments and Suggestions for Authors

The paper, titled "Unicompartmental Knee Arthroplasty for Osteoarthritis Eliminates Lateral Thrust: Associations between Lateral Thrust Detected by Inertial Measurement Units and Clinical Outcomes," investigates the relationship between clinical outcomes and lateral thrust before and after unicompartmental knee arthroplasty (UKA) using inertial measurement sensor units. It demonstrates a significant reduction in lateral thrust post-UKA, suggesting improvements in knee stability. Clinical outcomes, as measured by the Oxford Knee Score (OKS) and Japanese Orthopaedic Association Score (JOA), showed significant improvement postoperatively. The study concludes that UKA effectively eliminates lateral thrust in patients with osteoarthritis, contributing to better clinical outcomes and lower limb alignment.

This study provides valuable quantitative evidence supporting the efficacy of UKA in treating osteoarthritis patients with lateral thrust. It highlights the importance of using inertial measurement units for precise assessment of knee movement, offering insights into the benefits of UKA over traditional methods. Future research could expand on these findings by increasing the sample size and evaluating long-term outcomes to further validate the benefits of UKA in knee osteoarthritis treatment.

Comments on the Quality of English Language

The English language only have minor issues.

Round 2

Reviewer 1 Report (Previous Reviewer 1)

Comments and Suggestions for Authors

All comments have been addressed.

Comments on the Quality of English Language

no

This manuscript is a resubmission of an earlier submission. The following is a list of the peer review reports and author responses from that submission.

Round 1

Reviewer 1 Report

Comments and Suggestions for Authors

This manuscript entitled “Unicompartmental Knee Arthropla sty for Osteoarthritis Eliminates Lateral Thrust: Associat ions between Lateral Thrust Detected by Inertial Measurement Units and Clinical Outcomes” investigates the relationship between clinical outcomes and changes in lateral thrust before and after knee arthroplasty using inertial measurement sensor units (IMUs). In the reviewer's opinion, the methods and results sections of this article are vaguely described. Particularly in the Experimental Methods section, a more detailed, clear, and comprehensive description should be provided so that the reviewer understands the details of the gait analysis and the credibility of the results.

Specific comments are shown below:

Abstract:

1. The reviewer suggested that the author revise the abstract, which is an important part of attracting readers' interest. Concise and succinct explanation of the main results in the abstract, especially the section on the reduction of patients' preoperative lateral thrusts and the improvement in postoperative FTA/HKA.

Introduction:

2. The introduction is the beginning of the article that provides the background and context for the subsequent sections, and the conceptualization of the introduction as well as the logical aspects of its writing need to be strengthened. The reviewer did not fully comprehend or find the theoretical framework of the article to be particularly clear.

Methods:

3. In this article, Fig. 1 is presented unclearly, and the image specifications are provided.

4. Line 86: The information of the subjects is vague, the reviewer isn’t clear about what other information they have, and the gender of the subjects should be marked.

5. Lines 89–92: Why do you choose people around 40 years old, and what are their typical characteristics? Please quote the previous research. In addition, provide more details on the criteria for subject selection, including the extent of knee osteoarthritis to be included in the study and the definition of being able to walk without using a walking aid.

6. Lines 101–103: Give more information about the specific conditions and environment in which the walk-through analysis was conducted to ensure that the reader has a clear understanding of the experimental conditions.

7. Lines 116–117: When describing FTA and HKA, more detailed measurements could be provided, including the anatomical landmarks used and the specific steps of the measurements, to increase the transparency of the methodology. Emphasize why FTA and HKA were chosen to assess lower extremity alignment so that the reader can understand the importance of these two perspectives in the study.

Results:

8. Lines 130-131: More information about the observation period is provided to ensure that the reader understands how the average observation period of 9.1 months was selected and measured.

9. Lines 131-135: Use clearer language to explain the data when describing peak varus velocity, including the significance of the peak velocity and how it compares to cut-off values from previous studies.

10. Lines 140-142: The reviewer recommends emphasizing the significant improvements in OKS and JOA scores when describing clinical outcomes and discussing the practical impact of these improvements on patient quality of life and surgical outcomes.

Discussion:

11. Lines 193-194: As the author mentioned in the limitation part, there are big problems in the small number of cases, which is also the concern of the reviewer.

Conclusion:

12. Why does the experiment use IMU? Are there certain advantages to using IMU for quantitative assessment? If so, it is possible to summarize the discussion. Finally, the reviewer suggests summarizing the actual clinical significance of the study results, which would then echo the title of your article.

Comments on the Quality of English Language

 Moderate editing of English language required.

Reviewer 2 Report

Comments and Suggestions for Authors

The manuscript investigates the relationship between clinical outcomes and changes in lateral thrust before and after unicompartmental knee arthroplasty (UKA) using inertial measurement sensor units (IMUs). As a parameter, angular velocity of FTA and HKA was employed to determine the improvement of UKA. The results showed that the lateral thrust decreased to 55.6% after operation.

The proposed study is of significance in evaluating the effectiveness of UKA surgical interventions. However, for comprehensive coverage, certain aspects should be addressed.

1)      The authors measured only the angular velocity which was averaged in three-times measured. But the angular velocity was varied according to the walking velocity. Also, patient’s age, hospitalization duration, and weight would be affected to the result.

2)      In page 4, line 131, it is mentioned that nine out of eleven knees exceeded the cut-off value, and among these, five out of nine knees showed improvement after the UKA, falling below the cut-off value. Then, further investigation is warranted to elucidate the underlying factors contributing to the observed difference between the two groups.

3)      In page 3, line 100, the authors mentioned averaging the data of 10m walking performed three times. It would be more informative to provide details such as the number of steps considered in the averaging process, specifying whether the averaging is based on individual steps or extracted peak values.

4)      The graphical representation of walking velocity in three axes would improve visibility and comprehension.

Reviewer 3 Report

Comments and Suggestions for Authors

The entire manuscript is a comparison of the parameters of the anterior and posterior knee joints in patients with UKA. In fact, the meaning of the entire manuscript is very complete, and the introduction section also well summarizes why this research is done and the significance of doing this research. But there are big problems with the entire experimental part. 1. Selection of subjects. In the previous study, 8 subjects did not meet the statistical significance of the study. So how did the author arrive at his sample size calculation? 2. Why choose subjects over 40 years old? The age range of over 40 years old is too broad. In other words, the subjects selected by the author are not precise enough. 3. There are many methods for gait testing, such as footscan, vicon, etc. IMUs are indeed one of them, but in many studies I have seen, because of the limitations of the parameters derived from IMUs, IMUs are generally used as an auxiliary test. Furthermore, I think the results from IMUs do not allow us to very well deduce why we have the results we do, and the reasons for results. 4. The author should conduct tests at different times after surgery, such as 1 month, 3 months, and 6 months. 5. The author should control walking speed, because researchers can better control variables by controlling walking speed, especially after surgery such as UKA.
